# Electrospinning of Cellulose Nanocrystal-Filled Poly (Vinyl Alcohol) Solutions: Material Property Assessment

**DOI:** 10.3390/nano9050805

**Published:** 2019-05-27

**Authors:** J. Elliott Sanders, Yousoo Han, Todd S. Rushing, Douglas J. Gardner

**Affiliations:** 1Advanced Structures and Composites Center (ASCC), School of Forest Resources (SFR), University of Maine (UMaine), 35 Flagstaff Rd., Orono, ME 04469, USA; yousoo.han@maine.edu; 2Army Engineer Research and Development Center (ERDC), 3909 Halls Ferry Rd., Vicksburg, MS 39180, USA; todd.s.rushing@usace.army.mil

**Keywords:** poly (vinyl alcohol), cellulose nanocrystals, electrospinning, polymer nanocomposites, tensile properties, scanning electron microscopy, rheology, thermogravimetric analysis

## Abstract

Poly (vinyl alcohol) (PVA) and cellulose nanocrystals (CNC) random composite mats were prepared using the electrospinning method. PVA/CNC mats were reinforced with weight concentrations of 0, 20 and 50% CNC (*w*/*w*) relative to PVA. Scanning electron microscopy was used to measure the fiber diameter, which ranged from 377 to 416 nm. Thermogravimetric analysis (TGA) confirmed the presence of CNC fibers in the mat fibers which were not visible by scanning electron microscope (SEM). Mechanical testing was conducted using ASTM D 638 on each sample group at 10 mm min^−1^. Neat PVA and PVA/CNC mats were heat treated at 170 °C for 2h hours, and the morphological structure was maintained with some fiber diameter reduction. Mechanical property results after heat treatment showed a decrease in tensile strength, an increase in tensile stiffness and a decrease in strain to yield (%). This effect was attributable to enhanced diffusion bonding of the mat fiber intersections. The CNC fibers also increased mat stiffness, and reduced strain to yield in non-treated mats. The use of CNCs show potential for compounding into bulk polymer composites as a reinforcement filler, and also show promise for chemical crosslinking attributable to the –OH groups on both the PVA, in addition to esterification of the vinyl group, and CNC.

## 1. Introduction

Since the late 20th century, electrospinning has become a practically useful method for producing value added materials from dissolvable plastics, and more recently a method for producing nanocomposite materials [1]. Solution parameters influencing the electrospinning process include polymer concentration, solvent evaporation time, charge carrying capacity (conductivity), pH, average molecular weight (*M_w_*) and the degree of hydrolysis (DH) of the polymer. A universal theory was also outlined which details how entanglement molecular weight (*M_e_*) is useful for determining concentration levels capable of forming continuous electrospun filaments a priori. Shenoy et al. [2] utilized an equation which is used to calculate the solution entanglement number (${\left(n_{e}\right)}_{soln})$. This number describes at what point the product of polymer concentration or volume fraction ($\phi_{p}$) and molecular weight (M) of a polymer constitute fiber formation of an electrospun dope. Fiber production occurs, via intramolecular interactions, with approximately 2.5 per chain entanglements in polymer/good solvent systems.
(1)$${\left(n_{e}\right)}_{soln}=\frac{M_{w}}{{\left(M_{e}\right)}_{soln}}=\frac{\left(\phi_{p}M_{w}\right)}{M_{e}}$$

Factors of the electrospinning instrument include the needle tip to collector distance (TCD), the inner gauge diameter of the dispensing needle [3], the voltage intensity (kV) [4] and the pump feed rate. These parameters are outlined and comprehensively discussed in a review written by Haider et al. [5]. 

The practical focus of electrospinning is for development of filtration substrates: Such as for desalination [6] and air, water [7] or oil [8] filtration [9], nano-composite fiber reinforcements and for use as an interlayer in composite material systems like laminates and textiles. Electrospun fibers show potential in filtration because of the controlled fiber diameter, high specific strength, high surface area and mat pore sizes. However, the fiber network’s mechanical instability in comparison to cast films made with the same materials prevents its use in air and liquid filtration substrates. Internal filler orientation within the submicron fibers [10], and alignment of the submicron fibers [11,12] by collection on a rotating drum, are approaches that can increase the mechanical strength of a spun mat [13]. Submicron fiber size and controlled fiber length, in combination with a nano-scale reinforcing filler, provide material rigidity and the material properties can be measured with nano-indentation [14] before compounding. Chemical crosslinking of electrospun mats by physical (heat treatment) or chemical (solvent soaking) means improves the mechanical properties within the spun mat or membrane systems. Interfacial interactions and crosslinking with sizing agents (silane) and heat treatment could be used in reinforced matrix systems or interlayer systems. Further development in these areas is necessary.

Poly (vinyl alcohol) (PVA) is a hydrophilic polymer that dissolves in water. Its desirable properties include abrasion resistance, alkali resistance, gas permeability in nonwoven fiber network, oxygen barrier properties of films, and biodegradability. As the degree of hydrolysis (DH) increases, so do its strength, water resistance and chemical compatibility [15]. Early research conducted with electrospun fiber production focused on using 87–96% hydrolyzed PVA [4,15,16,17]. The surface tension of the aqueous solution increased as the DH in PVA increased, such that the electric field of the electro spinning mechanism was incapable of overcoming the surface tension at 99+ DH. From 87 to 95% DH, the surface tension increased from 51 to 54 mJ/m^2^ while water has a surface tension of 72 mJ/m^2^. A 99% hydrolyzed PVA (≈69 mJ/m^2^) can be spun into fibers, but only with the help of a surfactant at concentration levels greater than or equal to 0.5% (*v*/*v*). The addition of Triton X-100 surfactant reduces the contact angle of 100% hydrolyzed PVA from 102.2 ± 0.4° to 60° after using 0.3 *v*/*w* % [16].

In desalination, a novel hydrophobic/hydrophilic (1:4 ratio) of polyethylene terephthalate (PET) and PVA interpenetrating network support was produced for a forward osmosis membrane. Improvements to flux were attributable to the wettability and water transfer capabilities of the composite support layer [18]. The use of nanofibers in air and oil filtration substrates as an added electrospun nano-layer was studied by Feng et al. [8]. Submicron fibers produced pore sizes of 14.4 µm accuracy in the nano layer, with pressure drops increasing by 3 L/min. 

Customized production of electrospun fibers was studied as a bulk mechanical reinforcement in a study from Karimi et al. [19]. In this study a conventional wet lay-up method was compared to a newly developed electrospinning-electrospraying technique that utilized aligned PVA fibers as an epoxy reinforcement. It was noted that successful impregnation using vacuum consolidation and reduction of sprayed fiber handling produced results that created a shift in thermal properties, increased strength of the spun nanofiber epoxy, and increased modulus and elongation at break; results not typically seen in other nano-fillers. The increased interfacial adhesion between fiber and matrix were credited with the increased mechanical performance. Elongation improvements were proposed to be a result of percolation networks that redistributed mechanical failures in the matrix. The authors also noted that the process is scalable for automated processing.

For applications where a disturbance of spun fibers is likely, post treatments can be used to fuse fiber networks together and affect nanofiber mat properties like pore size, fiber diameter, hydrophobicity, elongation and tensile strength and stiffness. Es-saheb and Elzatahry [20] studied thermal post treatment, above and below the glass transition temperature (T_g_) on PVA sheets. The authors observed a strain rate dependence of the tested samples with mechanical properties increasing with the increased strain rate. Additionally, mechanical properties were improved with heat treatments at 70% of the melting temperature (T_m_). Guirguis & Moselhey [21] reported a T_g_ of 100% hydrolyzed 125 kg mol^−1^ PVA to be 209.6 °C. The melting temperature increases with increased DH and M_w_. Miraftab et al. [22] conducted a similar heat-treatment study with the additional comparison of a methanol treatment; a similar study was done before by Franco et al. [23]. It was reported that heat treatments did not affect fiber morphology, caused yellowing or discoloration, increased water stabilization, increased crystallization, increased fiber diameter, and produced mechanically strong and stable fibers at temperatures of 180 °C for durations up to 8 h. Methanol treatment coalesced fiber morphology, leaving some pores, and improved swelling when saturated with water. In the Franco et al. study, the addition of glutaraldehyde increased elongation. Wong et al. [24] studied fiber diameters and water absorbance effect on fiber diameter with respect to four different heat treatment temperatures, ranging from 85 °C to 160 °C, at 4 h lengths, followed by 1–30 day durations of water exposure. Fiber diameters increased with heat treatment, then decreased with exposure in water. The stability of fibers, and the increase in elastic modulus, was attributed to a minimum of 4 h of treatment at 135 °C; these properties were an effect of increased crystallinity. The author also noted that increased temperatures at shorter heat exposure times would yield similar results.

Cellulose nano crystals (CNC) can be derived from numerous sources and are typically created by acid hydrolysis of native cellulose, beginning with an acid and deionized (DI) solution, followed by mechanical shearing via a centrifugation or an ultra-sonication treatment to remove amorphous regions, and ending with a washing phase [25,26]. The dimensions of CNC produced from plant sources typically range from 100–700 nm in length (L) and 5–30 nm in diameter (D) [26,27]. These sizes are suitable for utilization in submicron fiber reinforcement. The elastic modulus of CNCs were theoretically estimated to 220 ± 50 GPa [28], and measured in cotton fibers with Raman spectroscopy at 105 GPa [29]. Lee and Deng [14] employed nano-indentation to measure both the modulus of pure PVA and 20 wt% cellulose nano-whiskers reinforced nanofibers. The moduli were 2.1 and 7.6 GPa, respectively. The authors reported a linear increase with the addition cellulose nano-whiskers up to 20 wt% content. Experimental results were 60–80% higher than the isotropic Halpi-Tsai model results, but lower than the longitudinal predictions; this suggested the nano-whiskers were partially oriented within the spun fiber. Dong et al. [10] reported a 17% increase in storage modulus using nano-intentation dynamic mechanical analysis (nano-DMA) with the addition of 17 wt% CNC.

Thermal degradation of CNC begins near 200 °C, and is measured using thermogravimetric analysis (TGA) [26]. More detailed degradation data that was based on the drying method was observed and reported by Peng et al. [30]. Micro-crystalline cellulose (MCC) degraded in three steps, and was characterized by dehydration in regions 1 and 2, followed by chain depolymerization and breakdown into hydrocarbons in stage 3. Addition of CNC created a thermal shift via –OH interactions with the composite matrix. Dong et al. [10] reported a thermal shift that was attributable to hydroxyl interactions with the carbonyl groups in the poly (methyl methacrylate) matrix. Such shifts were reported in PVA/CNC electrospun nanofibers, and were attributable to hydrogen bonds between PVA side chains and CNC [13].

Cellulose and PVA composites show potential for material development due to the combined light-weight, bio-functional and environmentally friendly properties. Meree et al. reported mechanical properties with a 13% increase in strength and a 34% increase in modulus, with the addition of 3% (*w*/*w*) microcellulose. Additionally, softwood Kraft pulp has shown a doubling of tensile strength, and 2.5 times increase in the stiffness of a PVA matrix with the addition of 5% (*w*/*w*) [31]. 

In an effort to better understand the viscoelastic behavior of highly loaded PVA/CNC composite suspensions, rheological measurements were performed to study the composite behavior for applications requiring water-based processing. CNC loading levels of 20–67% (*w*/*w*) were combined with PVA that had an M_w_ of 31–50k, up to 146–186k. The increased M_w_ of PVA showed an increase in crystallite formation, elasticity and viscosity. As the solution aged, the rate of aging, defined by the entanglement of polymer chains in a still solution, decreased with increased time. The CNC was shown to reduce the kinetic driving force between the stratification of PVA and water in solution. Additionally, the percolation threshold of CNC had an effect on the storage modulus (G’), in that at a lower M_w_ of PVA, the intramolecular entanglement played a larger elastic role as its concentration increased. However, at a high M PVA, the percolation threshold attributable to CNC is the contributing factor in rheological responses. In solution, CNC was known to show shear thinning behavior [25]. At 7% (*w*/*v*) concentration of PVA with 5, 10 and 15% (*w*/*w*) CNC the G’ was increased three times, using fully hydrolyzed PVA (99%) [17]. A lesser hydrolyzed polymer network was expected to show lower mechanical property values because of less intermolecular hydrogen bonding within the random electrospun network. 

This body of work attempted to synthesize the cited literature into a repeatable method with the intent to produce consistently manufactured composite PVA/CNC nanofibers with electrospinning techniques. This study is unique, in that a highly filled PVA/CNC dope was spun and characterized by visual, thermal and mechanical means. Additionally, a post-processing treatment was utilized to investigate the heating effect on fiber morphology and mechanical properties. The target applications are for the use of these fibers as filtration media, fiber reinforcement or interlayer composite structures.

## 2. Materials and Methods

### 2.1. Sample Naming Conventions

The poly (vinyl alcohol) (PVA) concentration in distilled water was constant at 7% (*w*/*v*); therefore, it was not included in the labeling. The PVA within the dry mat was labeled as 100_PVA_ for neat mats, 80_PVA_ for 20% cellulose nanocrystals (CNC), and 50_PVA_ for 50% CNC (*w*/*w*). Similarly, the CNC content in dry mats were expressed as 20_CNC_ or 50_CNC_. In combination, a neat PVA mat is coded 100_PVA_ and a 20% (*w*/*w*) CNC mat is coded 80_PVA_20_CNC._ Heat treated samples were delineated from as-spun samples with an H label after the CNC content. For example, a neat PVA mat with heat treatment was expressed as 100_PVA_H, whereas an as-spun 50% (*w*/*w*) mat was expressed as 50_PVA_50_CNC_.

### 2.2. Solution Preparation

A 99% hydrolyzed poly (vinyl alcohol) (PVA) with an M_w_ of 130,000 g mol^−1^ was purchased from Sigma Aldrich (St. Louis, MO, USA). PVA suspensions containing aqueous cellulose nanocrystals 12.5% (*w*/*v*) (CNC), produced by the USDA Forest Products Laboratory (FPL) (Madison, WI, USA), were dissolved in de-ionized (DI) water at approximately 90 °C for 3 hours in a silicone oil bath under magnetic stirring. DI water was chosen as not to increase processing complexity, and because the water evaporated upon fiber collection without affecting fiber morphology. Fiber dimensions were reported by the manufacturer at 5–20 nm in width and 150–200 nm in length [32]. Peng et al. [33] performed scanning electron microscope (SEM) analysis and particle size distribution of CNC; see the first figure (sections g–i) of the referenced material. The solution was kept under magnetic stirring during cooling to room temperature. After the solution was cooled, the surface tension was reduced for electrospinning by adding 0.5% (*v*/*v*) Triton X-100 M_w_ 80,000 (g mol^−1^), Sigma Aldrich (St. Louis, MO, USA) surfactant. Solutions were prepared in the quantities of 7% PVA (*w*/*v*) and 0, 20 and 50% (*w*/*w*) CNC. The CNC weight was based on the dry basis of PVA. Solutions were stored in a sealed vessel containing a beaker of DI water to reduce the dehydration rate of the solutions. Upon using this storage method, solutions were kept for 2+ weeks without biological contamination. Solutions were magnetically stirred before electrospinning.

### 2.3. Electrospinning Parameters

Electrospun mats were produced with a NanoNC (Seoul, Korea) eS-Robot Electrospinning/spray System (Model ESR200R2P) using a single phase, 220 V, 60 Hz, with a voltage maximum of 30 kV. Th solution was dispensed through a 25 mm-long 18 G, 304 stainless steel blunt tip reusable needle, with 0.0472” outer diameter and 0.0315” inner diameter (IntelliSpense; Agoura Hills, CA, USA) that served as the anode. A rectangular flat plane glass collector was used for the grounding and collection substrate. Adhered to the collector, with 3M double-sided polypropylene tape, was a 22.86 × 38.1 cm sheet of 304 stainless steel, provided by NanoNC. A polytetrafluoroethylene (PTFE) sheet was cut with a 22.86 × 38.1 cm inner dimension and a 7–10 cm outer margin to direct the electric field to the collection area, effectively insulating the outside from collecting fibers. The robot feature on the machine was programmed to move at a rate of 50 mm/min along the lengthwise axis (*x*-axis), and incrementally step down 5 mm after each 70 mm pass within a 55 cm height (*y*-axis). For each mat, 1.86 mL of solution was dispensed, and the TCD was 15 cm. The TCD was determined based on precedent literature. This distance was chosen so that fiber morphology formed as opposed to a film [4,34,35]. Santos and Elchorn selected a TCD after running their own trials and selecting for best electrospun fibers [35]. The conditions, such as supplied voltage and feed rate, are changed relative to cellulose content in attempts to maximize throughput without creating droplet defects like those seen in Figure 1c,d. Mats were prepared at a 19–20 kV and 24.5–34 µL min^−1^ feed rate, as referenced in Table 1.

### 2.4. Scanning Electron Microscopy

Scanning Electron Microscopy (SEM) was performed using a Hitachi 2000 Tabletop Microscope SEM (Tokyo, Japan). SEM micrographs were used for fiber diameter analysis. Measurements were taken using the software ImageJ (National Institute of Health), and the measurement tool was scaled to the 10 µm scale bar of each 10,000× zoom micrograph (see images in Figure 2). Forty measurements were taken in each micrograph over three separate mats composed using the same electrospinning parameters and then averaged. One micrograph of the three mats used for measurements for each sample set is shown in Figure 1.

### 2.5. Viscosity and Rheology

Viscosity and rheology measurements of PVA and PVA/CNC solutions were performed using a Bohlin Gemini parallel plate (25 mm) rheometer (Malvern Instruments, UK). A shear ramp at room temperature (23 °C), using shear rates ranging from 0.1 to 100 (1/s), was used to measure viscosity (Pa∙s). A small amplitude oscillation test was performed with a strain sweep from 0–200% to measure the linear viscoelastic regime of each solution. A strain amplitude of 1% of the elastic modulus was chosen for frequency sweep measurements. The elastic modulus (G’), storage modulus (G”) and complex viscosity (η*) were measured with a 0.1–100 hertz (Hz) frequency sweep. Large phase angle measurements were omitted from the reported data due to measurement lag, resulting in instrument signal noise.

### 2.6. Thermogravimetric Analysis (TGA)

Measurements were taken with a TA Instruments Q500 (New Castle, DE, USA). Approximately 10 mg of mat sample was used for each measurement. Samples were tested within a 30 °C to 600 °C temperature ramp with an increasing rate of 10 °C min^−1^ under nitrogen atmosphere (20 mL min^−1^) to avoid oxidation. 

Tensile testing, using a universal testing machine (Instron 5966) with a 10 kN load cell at 10 mm min^−1^ was performed using ASTM D-638 specimen IV dimensions. Three to four mats were produced for each sample category, and two samples were cut and tested from each mat. The average of the width (w), thickness (d), tensile strength and modulus were recorded. Grip surfaces were covered in adhesive paper. Samples were not conditioned, but were tested within 24 h of preparation. Heat treated samples were compressed between two PTFE sheets followed by two 0.32 cm thick aluminum sheets, and topped with a steel weight for a 2 h period at 170 °C then stored at 50% relative humidity and 23 °C.

## 3. Results and Discussion

### 3.1. SEM and Fiber Diameter

The average measured fiber diameters are reported in Table 2 and were within a range similar to cited literature with similar 99% DH and 8–15% PVA (*w*/*v*) solution concentrations [14,15,36]. The largest and smallest average diameters measured were 50_PVA_50_CNC_ and 80_PVA_20_CNC_ samples at 456 ± 85 nm and 358 ± 80 nm, respectively. The maximum and minimum measured fibers were observed in the 100_PVA_ and 80_PVA_20_CNC_ sample groups at 692 nm and 233 nm, respectively.

Authors Lee and Deng [14] reported that fiber diameters decreased with decreased kV (10–25 kV). Fiber diameter also decreased with increasing cellulose nano-whisker content (wt.%); the TCD was reported to be 25 cm. The decrease with increasing CNC content observed in all CNC filled groups, with the exception of 50_PVA_50_CNC_, coincides with the aforementioned literature. A 20 kV voltage was used in this study for numerous reasons. Primarily, a large enough voltage was needed to overcome the surface tension of the spinning dope. At lower voltages, a droplet formed between the needle tip and Taylor cone. If unaffected by the electric field the droplet fell onto the spun mat. This effect was observed for a wide range of feed rates. Unspun droplets created spot defects in the mats (Figure 1d), and potentially affected elongation properties. Secondarily, the PVA sheathe covering the CNC may have thinned as the fiber diameter decreased with heat treatment. Additionally, CNC agglomeration, or an increase in the internal filler diameter, may have further reduced the outer layer of the submicron fiber with the decrease in diameter. This phenomenon was explained in the review from Wang et al. [37].

A heat treatment regime below the T_m_ of PVA, but above the T_g_, caused spun fibers to fuse at intersected points [20,23]. The fibers were visible in SEM micrographs, and intersections were visible as higher contrast white areas, shown in Figure 2. An especially large aggregation of randomly aligned fibers, fused together during heat treatment, is visible in the center of Figure 2d. Discoloration or yellowing was observed as a result of thermolysis [22]. The fiber diameters decreased when compared to the as-spun groups in both the 100_PVA_ and 50_PVA_50_CNC_, but not the 80_PVA_20_CNC_ groups; these data are reported in Table 2. The fibers did not appear more flattened or fused than the untreated mats. The preservation of fiber morphology shown in Figure 2 is consistent with Miraftab et al. [22]. Fiber diameter reduction of 10% was reported after 4 h at 135 °C in the Wong et al. [24] study. Given similar treatment conditions, the 50% (*w*/*w*) CNC group results reported in this study deviated slightly from the 10% reduction reported in literature, but the other groups coincided with a deviation of a few percentages. No CNC were visible at the 10,000× magnifications of SEM micrographs with a 10 µm scale bar.

### 3.2. Viscosity

A shear ramp at rates between 1 and 100 (s^−1^) was applied on electrospinning dopes to obtain instantaneous viscosity (η) data (Pa∙s) that are shown in Figure 3. The solution viscosity was 0.9 to 1.5 Pa∙s, and exhibited similar viscosity to that reported elsewhere at high M_w_ (145–185 kg mol^−1^) and with high levels of CNC filler [31]. Enayati et al. reported 209 cP at 8% (*w*/*v*) of PVA with a 98.9% DH and M_w_ of 125 kg mol^−1^ [36]. The higher viscosity response seen in the results shown in Figure 3 may be attributable to the use of parallel plates, rather than cone and plate geometries. Marginal differences in concentration and PVA M_w_ may have also played an additive role in the increased viscosity reported in this study. 

Both PVA and CNC showed shear thinning behaviors in aqueous solutions and suspension, depending on concentration. With CNC, at higher concentrations and shear rates between 0 and 1 s^−1^, there was an initial increase in viscosity attributable to fiber network interactions. After the physical alignment of the CNC network, a more pronounced drop from 10 Pa∙s was observed in 50_PVA_50_CNC_ samples after shear rates exceeded 0.2 (s^−1^) [25]. In the case of PVA, hydrogen bonding of hydroxyl (–OH) groups was a major mechanism promoting shear thinning, and was affected by increased M_w_ of polymer chains. Water and crystallites that formed by polymer chain interactions with the water solvent in standing solutions affected viscosity after 5 days [31]. 

Viscosity was important to measure because of its effect on fiber morphology during spinning. High viscosity may reduce pumping capacity or render the dope unable to be spun. With a shear thinning material, an increased pump pressure may cause sudden deposition of spinning dope as a result of shear forces increased inside the syringe; see Figure 1c. At high enough pressures the viscosity of the solution was decreased, and ejection of the solution occurred until the pressure was reduced. One solution to this processing issue, used in this study, was increasing the diameter of the spinning needle. Another approach was to use the relationship, presented by Shenoy et al. [2], to determine, a priori, the conditions (e.g., M_w_) that produce fibers. Relating viscosity data for high levels of CNC to the morphology of the spun fibers is advantageous for future research of these materials.

### 3.3. Storage and Loss Moduli

Frequency sweep results of the storage modulus (G’), loss modulus (G”) for 100_PVA_, 80_PVA_20_CNC_, and 50_PVA_50_CNC_ suspensions are shown in Figure 4. In the 100_PVA_ samples, the G” was greater than the G’, signifying the tested solution exhibited more liquid-like characteristics than solid gel-like characteristics. Additionally, the reduction of slope in the storage modulus as the CNC loading level increased suggested that the elastic dependence on frequency of G’ decreased with the increased CNC filler [31]. The attributable factor is related to the CNC percolation threshold, lack of PVA chain and crystallite formation interactions. According to the 50_PVA_50_CNC_ data, the suspension did not reach gelation state despite the superposition of G’ and G” responses. These data coincided with the feed rate data reported in Table 1. Lower feed rates were necessary as the electrospinning dope reached more gelatinous states. However, spinning was possible with the aid of the Triton X-100 surfactant.

### 3.4. Complex Viscosity

Complex viscosity (η*) was measured in a frequency sweep along with the G’ and G” and is shown in Figure 5. The response of CNC is clearly visible between the two 80_PVA_20_CNC_ and 50_PVA_50_CNC_ and the 100_PVA_ sample groups. The neat samples showed an independence in response to increased frequency (Hz). The increase in η* in was attributable to the CNC domains acting in conjunction with the liquid crystalline interphase of the PVA dope. These domains resisted forces during the frequency sweep and resulted in some dependence as frequency increased [31]. Conversely, once the PVA in the dope no longer acted as the primary resisting component, the 50_PVA_50_CNC_ sample group showed an increase in η* compared to 80_PVA_20_CNC_ and 100_PVA_. A decline was observed afterwards with the increased frequency. As the CNC networks were broken or weakened, the resistance decreased as a response to the increase in frequency, and met with 80_PVA_20_CNC_ samples near 1 Hz.

### 3.5. Thermogravimetric Analysis

Neat PVA and PVA/CNC mats were degraded via TGA, and the weight (%) and derivative weight loss (°C/%) data are shown in Figure 6 and Figure 7, respectively. The neat data showed a distinct mass loss that was attributable to water starting below 100 °C in both figures. The decomposition of PVA followed a two-stage process. The onset of PVA degradation occurred at 175 °C, and continued in the range of 240–350 °C, which agrees with non-thermally treated, 99% DH results from the reported literature. The first significant stage occurred at 240–340 °C and the second at 390–470 °C [38]. The maximum rate loss (%/°C) occurred at ≈216 °C, dipped slightly and was sustained until 300 °C. This loss was attributable to PVA chain stripping and chain scission. The final loss was attributable to chain decomposition and volatilization of residual elements of the polymer backbone [17,20]. 

Cellulose degradation occurred in three stages. Freeze dried and spray dried CNC sourced from the FPL was characterized and reported by Peng et al. [30]. Results reported in the Peng et al. study should exhibit an accurate representation for the TG and dTG information regarding CNC material used in this study. The first region ranged from 25 °C to 216 °C, and the majority of mass loss was attributed to water evaporation. The second region ranged from the onset of thermal degradation at the end of the first to the dehydration of cellulose, occurring between 200–280 °C, followed by the de-polymerization of cellulose, occurring between 280–340 °C, with some simultaneous overlap with dehydration. Region three occurred at 358 °C and after 500 °C, where the levoglucosan decomposed into hydrocarbons and hydrogen, leaving char residue at the final temperature of 600 °C.

Samples containing PVA and CNC decomposed similarly to neat PVA, but the presence of CNC shifted the peak mass loss rate (%/°C) to higher temperatures. In electrospun poly (vinylidene fluoride) fibers, studied by Zhang et al., a similar shift was observed in the 250–330 °C cellulose decomposition region in the TGA data [39]. The gradual rate increase after 125 °C with a sharp increase at ≈200–250 °C were attributable to the free water loss of CNC and the onset chain stripping of PVA. Two distinct peaks of mass loss were then visible afterward in 50_PVA_50_CNC_, but appeared as one peak with a slight shoulder in the 80_PVA_20_CNC_ samples. This combination effect, attributable to hydrogen bonding interactions, showed a more combined effect with the PVA at the 20% (*w*/*w*) CNC loading level. However, for 50_PVA_50_CNC_ the first peak, shown in Figure 6, began at ≈216 °C, and dipped significantly at ≈290 °C. In this region, dehydration of CNC bound water and the de-polymerization of CNC occurred during thermal decomposition [25,30]. A second peak was attributable to CNC dehydration/de-polymerization, coupled with the chain scission of PVA, and occurred near 340 °C. The rate of decomposition slowly declined until 400 °C. The rate loss increased afterward and peaked near 500 °C. The final rate increase was attributable to the breakdown and volatilization of CNC and PVA backbone structures.

### 3.6. Mechanical Properties

Dimensional analysis, tensile strength and modulus of elasticity data are shown in Table 3, and were reported with mean and coefficient of variance (COV). The COV was above 20% for all of the strength and modulus data. Dimensions were under 20% COV in all samples except the 50_PVA_50_CNC_ thickness results. The low COV in dimensional analysis indicated that the electrospun production method, with high percentages of CNC (*w*/*w*) filler, was repeatable. The assumption in mechanical property variance was mostly attributable to the materials performance with internal defects creating variance rather than the dimensional instability of the tested specimens. Stress strain curves are shown by sample group in Figure 8. 

Tensile strength results in this study were similar to a study, using 0.1% PVPA and 12% (*w*/*v*) PVA with a 2, 10 and 24 h 150 °C post-heat treatment, performed by Franco et al. [23]. The tensile strength among groups declined with the addition of CNC and after heat treatment in all sample groups. Stress-strain curves are shown in Figure 8. This reduction in strength could be attributable to the overfilling of the interior of PVA fibers. Particle size played a large role in composite performance and affected stress concentration in the matrix. With increased filler content, a decrease in tensile strength is observed (see Figure 8a,c,e). The effect may be especially prevalent when spun fiber diameters are 300–400 nm in width, and the filler diameter was over half of that dimension [37]. In this particular case, if the CNC filler that was 5–30 nm in width, and then agglomerated within the core of the PVA matrix fiber, the combined widths of the CNC quickly became a significant portion of the electrospun fiber’s internal width. The agglomerated CNC diameter would then reduce the tensile strength by reducing the active surface area available for interfacial interactions like hydrogen bonding. The largest averaged tensile strength was observed in 100_PVA_ at 6.27 MPa and the lowest was 1.77 MPa in 50_PVA_50_CNC_ mats. After heat treatment the average tensile strength of all sample groups decreased. The largest decline, a 32% difference, was observed in 100_PVA_H mats. This loss was attributable to the fiber intersection fusion. This fusion of intersecting fibers significantly reduced the strain (%), see Figure 8a and b. With the mechanical testing rate dependence of spun mats and decreased strain at yield, lower tensile strength values at the same cross head test speed were observed. Similar results were reported by Es-saheb et al. [20]. The tensile strength increased between 50_PVA_50_CNC_ and 50_PVA_50_CNC_H mats by 15%. This is attributable to the diffusion of fibers intersections and the inability of the fibers to be drawn. This, however, does not imply that the fiber was an effective strength reinforcing agent in the PVA matrix, since the 50_CNC_ values were significantly lower than the neat PVA.

Modulus of elasticity data are shown in Table 3, and the mean and COV are reported. Standard deviation was over 20% in all sample groups, but were lowest in 100_PVAC_H and 50_PVA_50_CNC_ sample groups; these data are not shown because of the high COV displayed in each sample group. The correlation of a modulus of elasticity increase with increased CNC content (*w*/*w*) was less distinguishable between 100_PVA_ at 371.6 MPa, when compared to 80_PVA_20_CNC_ samples at 300.7 MPa. However, the 785.4 MPa modulus of elasticity in 50_PVA_50_CNC_ samples increased 211% with the addition of CNC. Defects in 100_PVA_ samples were attributable to droplets falling onto the spun mat. The unspun dope diffused the fibers, and may have promoted an artificial stiffening of samples, as evidenced by the shortened elongation of two samples; see the stress-strain curves shown in Figure 8a. At a 20% fiber loading level, CNC showed hydrogen bonding, as evidenced by the single dTG degradation peak seen in Figure 7. However, tensile stiffness was not increased as a result. The lack of hydrogen bonding may be attributable to decreased crystallization of PVA with the addition of additives [36]. The crystalline domain of the PVA interacting with the CNC may have created discontinuity within the spun fibers and decreased stiffness [10]. A stronger response was seen in the 50_PVA_50_CNC_ samples, and stiffness increased as CNC may have created a percolation effect within the spun fiber. 

With regard to heat treatment, in 100_PVA_H samples the tensile stiffness was highest at 1656.1 MPa in 100_PVA_. Fusion between fibers and molecular entanglement are factors contributing to the increase in stiffness. Strain (%) reduction and increased modulus of the mat appeared to be correlated, as seen in stress-strain graphs shown in Figure 8. The second strongest modulus of elasticity was 1570.4 MPa in 50_PVA_50_CNC_H, a 5% decrease from 100_PVA_. The crystallization and diffusion of PVA fibers coupled with half CNC (*w*/*w*) filler increased the 50_PVA_50_CNC_H stiffness. The 5% reduction, compared to 100_PVA_, may be attributable to a discontinuity of the matrix attributable to clogging during processing or the lack of PVA CNC adhesion described earlier.

Stress-strain data is reported in Figure 8. The higher strain percentages were attributable to a lengthening of the random mat and an aligning of the fibers during drawing; see Figure 1a,b. Fusion of the electrospun layers occurred as a result of the liquid droplet defects. The fused areas created stress concentration points and prohibited drawing of fibers under tensile load. A circular shape is visible in the middle of the sample shown in the specimen pictured in Figure 1d. This phenomenon effectively shortened the extension potential of the mat, and resulted in variation among other 100_PVA_ samples. This shortening effect is visible in the stress strain curves of 100_PVA_ and 80_PVA_20_CNC_ in Figure 8a,c, respectively. Reduction in drop defects may reduce the strain defects seen in decreased strain (%) samples. 

## 4. Conclusions

This work concludes that electrospinning of highly filled (up to 50% *w*/*w*) PVA/CNC composite material was possible with the chosen electrospinning parameters. Consistent morphology between as-spun mats was observed in comparison to heat treated sample groups by SEM. The fiber diameters of electrospun random mats were decreased in 100_PVA_ and 50_PVA_50_CNC_ by 12 and 18% as a result of PVA crystallization during heat treatment. The presence of CNC fibers in random PVA electrospun mats were confirmed by an increased thermal stability in 80_PVA_20_CNC_ and 50_PVA_50_CNC_ sample groups, increasing the materials’ thermal use range. The 80_PVA_20_CNC_ sample group decomposed in a homogenous fashion, while the 50_PVA_50_CNC_ decomposed into a two peak, three phase, pattern. The former degradation pattern was attributable to improved hydrogen bonding. The modulus of elasticity of 100_PVA_ mats increased by 211% as a result of 50% (*w*/*w*) added CNC filler, and may be attributable to percolation networks not observed in 20% (*w*/*w*) CNC stiffness results. This increase in tensile stiffness of as-spun submicron fibers could be applied as a means to fiber reinforce a polymeric resin matrix. Additionally, in comparison to as-spun samples, heat treatment of mats for 2 h at 170 °C increased tensile stiffness by 4.4-, 2.3- and 2-fold in 100_PVA_H, 80_PVA_20_CNC_H and 50_PVA_50_CNC_H, respectively. Decreased tensile strength and shortened strain at yield were observed with the addition of increased CNC content and heat treatment. In PVA and PVA/CNC sample groups the heat treatment promoted physical diffusion in the fiber mat and the increase in stiffness without noticeable morphological changes. These attributes could be utilized in combination with hydrophobic filtration media, and employed as an interlayer reinforcement of composite laminate materials.

## Figures and Tables

**Figure 1 nanomaterials-09-00805-f001:**
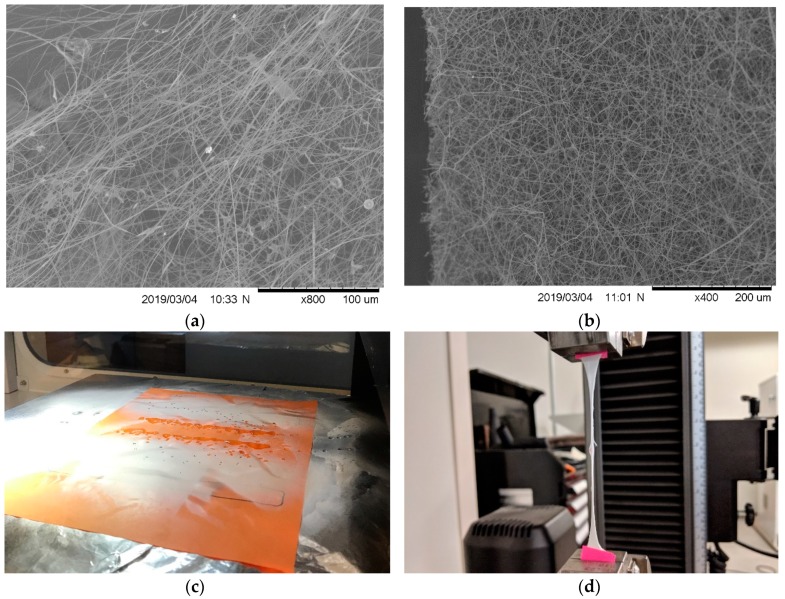
(**a**) 50_PVA_50_CNC_ drawn specimen taken by SEM after tensile testing, (**b**) the 50_PVA_50_CNC_ mat before mechanical testing, (**c**) pictured is an example of solution dispensing onto the grounded collector and producing defect in the mats, (**d**) an example of a circular film like defect produced from the dripping of solution onto the collector.

**Figure 2 nanomaterials-09-00805-f002:**
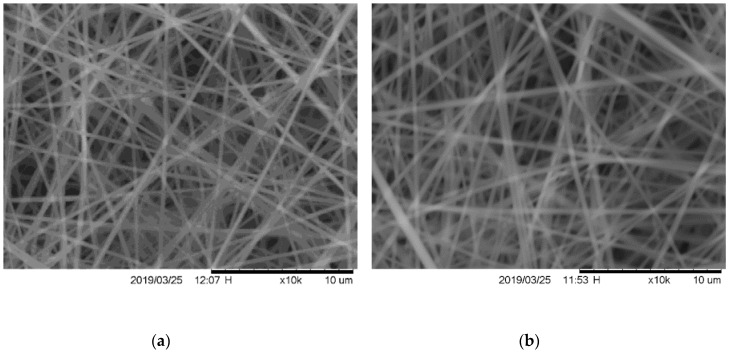
Scanning electron micrographs (SEM) at 10,000× magnification of random electrospun Poly (vinyl alcohol) (PVA) and PVA/cellulose nanocrystals (CNC) regular (left) and heat treated (right; H) mats. Items (**a**,**b**) are 100_PVA_ mats, (**c**,**d**) are 80_PVA_20_CNC_ and (**e**,**f**) are 50_PVA_50_CNC_.

**Figure 3 nanomaterials-09-00805-f003:**
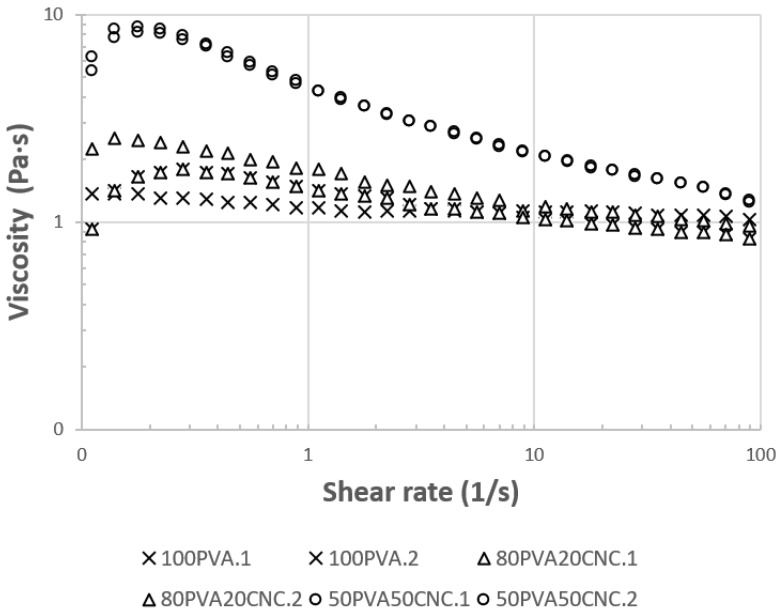
Instantaneous viscosity measured by a shear ramp from 1–100 (1/s). Two tests were conducted per sample type to ensure precision.

**Figure 4 nanomaterials-09-00805-f004:**
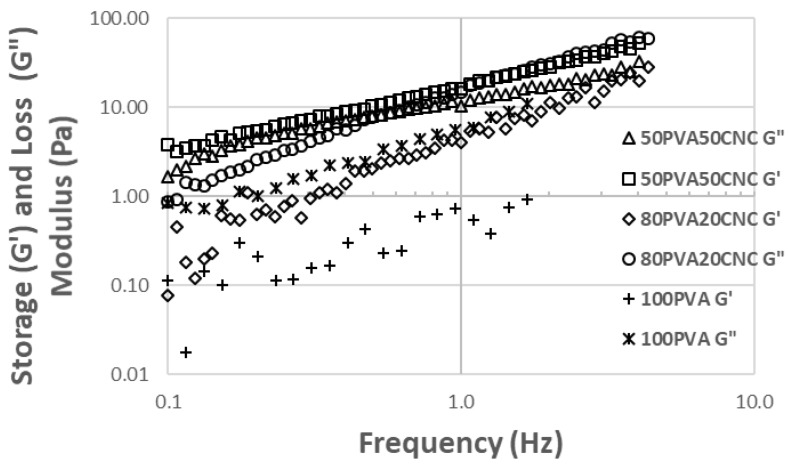
Data measured from a frequency (Hz) sweep (0.1–10) of aqueous suspension neat PVA and PVA/CNC at varying weight percentages. Some data was omitted because the phase angle was too large.

**Figure 5 nanomaterials-09-00805-f005:**
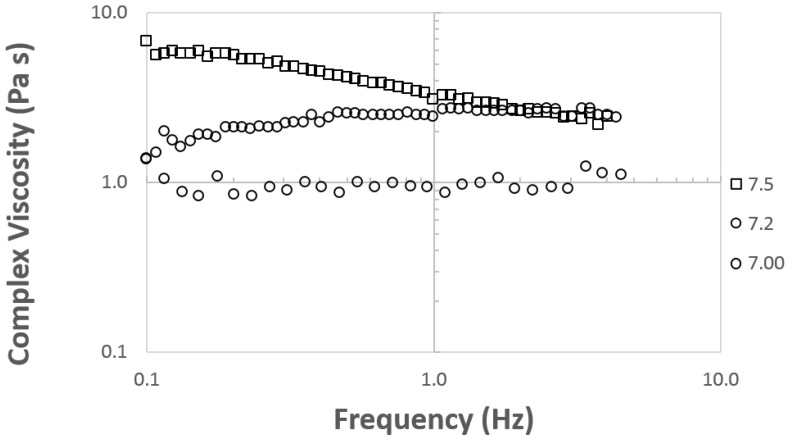
Complex viscosity data measured in a frequency sweep (0.1–10 Hz).

**Figure 6 nanomaterials-09-00805-f006:**
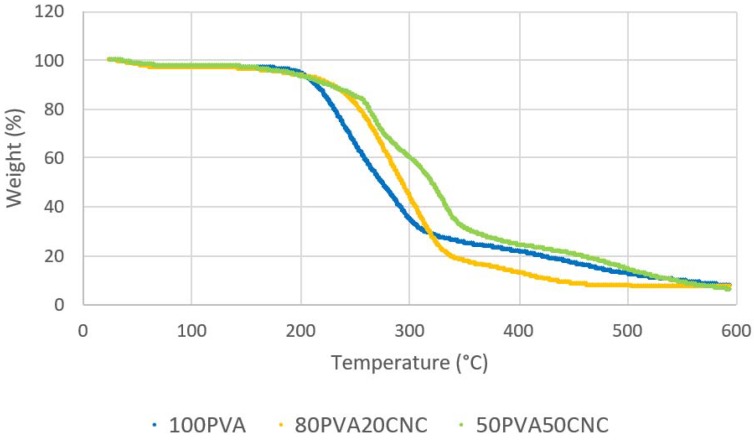
Thermogravimetric analysis (TGA) curves showing weight loss (%) of 100_PVA_, 80_PVA_20_CNC_ and 50_PVA_50_CNC_ electrospun random mats.

**Figure 7 nanomaterials-09-00805-f007:**
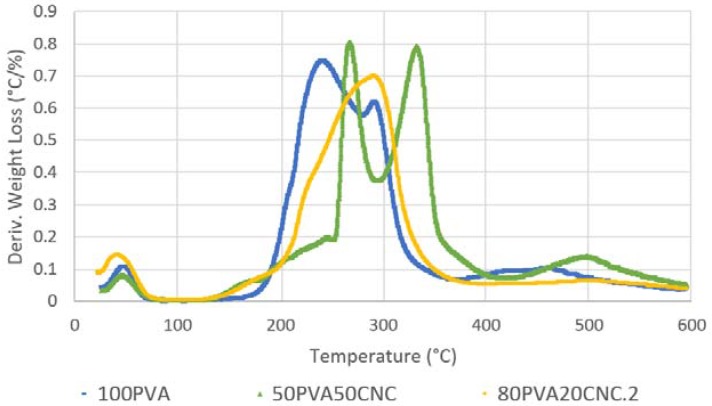
TGA curves showing derivative weight loss (%/°C) 100_PVA_, 80_PVA_20_CNC_ and 50_PVA_50_CNC_ electrospun random composite mats.

**Figure 8 nanomaterials-09-00805-f008:**
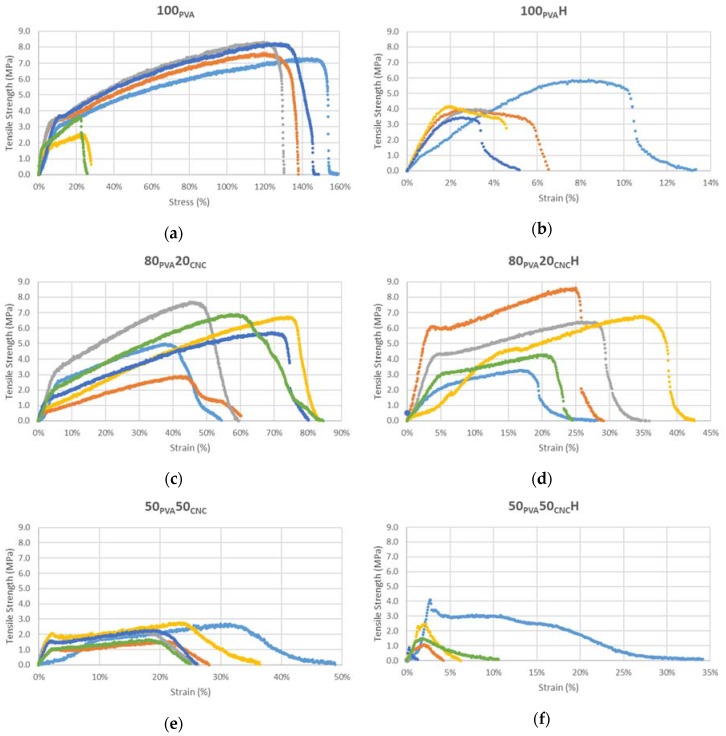
Stress-strain curves for 100_PVA_ (**a**,**b**), 80_PVA_20_CNC_ (**c**,**d**) and 50_PVA_50_CNC_ (**e**,**f**) tensile strength samples. Each sample group has six samples (with the exception of 100_PVA_), and heat treated samples are labeled with an (H). Colors are used to differentiate between individual samples *within* each group category.

**Table 1 nanomaterials-09-00805-t001:** The voltage supplied (kV) for the electrospun mats, the conductivity (µS/cm) of each solution and feed rate (µL min^−1^) for each mat type. The sample code is explained in Section 2.1.

Sample Code	Voltage (kV)	Conductivity (µS/cm)	Feed Rate (µL min^−1^)
100_PVA_	19–20	926.4 ± 3.1	34.0
80_PVA_20_CNC_	19–20	913.3 ± 3.3	31–32
50_PVA_50_CNC_	19–20	1037.8 ± 26.0	24.5

**Table 2 nanomaterials-09-00805-t002:** Average fiber diameter (nm) with standard deviation, minimum and maximum values by electrospun mat composition. The electrospun mats with heat treatment are labeled (H).

Fiber Diameter (µm)				
Sample	Mean	Stdev	Min	Max
100_PVA_	0.442	0.085	0.297	0.692
100_PVA_H	0.390	0.077	0.254	0.586
80_PVA_20_CNC_	0.358	0.080	0.233	0.593
80_PVA_20_CNC_H	0.408	0.074	0.271	0.575
50_PVA_50_CNC_	0.456	0.078	0.272	0.632
50_PVA_50_CNC_H	0.373	0.091	0.204	0.607

**Table 3 nanomaterials-09-00805-t003:** Tensile mechanical properties for electrospun PVA (7% *w*/*v*) and PVA/CNC composite mats with and without heat treatments. The mean and coefficient of variation are reported for all three types (00, 20, 50% *w*/*w*) and for no treatment and heat treatment (H).

	Thickness (mm)		Width (mm)		Modulus (MPa)		Tensile Strength (MPa)	
Sample	Mean	COV	Mean	COV	Mean	COV	Mean	COV
100_PVA_	0.09	9%	6.11	2%	371.6	35%	6.27	40%
100_PVA_H	0.09	16%	6.08	2%	1645.1	29%	4.26	22%
80_PVA_20_CNC_	0.09	15%	6.41	2%	300.7	51%	5.28	30%
80_PVA_20_CNC_H	0.09	25%	6.10	4%	690.7	70%	5.08	28%
50_PVA_50_CNC_	0.08	22%	6.12	2%	785.4	56%	1.77	32%
50_PVA_50_CNC_H	0.085	10%	6.25	2%	1570.4	37%	2.04	59%

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
