# Peer review of "Electrospinning of Cellulose Nanocrystal-Filled Poly (Vinyl Alcohol) Solutions: Material Property Assessment"

_nanomaterials, 2019, doi:10.3390/nano9050805_

Reviewer 1 Report

I’ve read the submitted paper that deals with the electrospinning of cellulose nanocrystal-PVA solutions and the evaluation of the resulting mat properties. I think this work is potentially interesting but, before being considered for publication, this manuscript should undergo some revisions:

1.       Introduction: should be edited removing the paragraph and linking the different topic presented in order to lead the reader through the literature and towards the aim of the paper. Also some section with detailed description of materials properties (e.g. nanocrystals degradation) should be avoided.

2.       The goal of the manuscript should be clearly stated in the introduction

3.       The novelty of the work should be clearly discussed with respect to the mentioned references.

4.       Figure 1: it would be nice to see micrographs at higher magnification.

5.       TGA: the TG and dTG of neat CNC should be reported. The authors should evaluated the residues of the two components at high temperatures in order to see if synergistic or simply additive effects occur during degradation (the use of calculated vs experimental tg plot is suggested). The weight loss of sample 7.20 in the 350-500 range is strange with respect to the other samples can the authors explain this ?

6.       The discussion section is more like a short review paragraph of other works and should be adjusted so that the main results of the paper a discussed and comparison with other paper is performed.

7.       The conclusion paragraph should be ameliorated as it only contains a summary of the results achieved. 

Author Response

J. Elliott Sanders

Nanomaterials Reviewer Responses

Dear Editor

Response to Reviewer 1 Comments

Point 1: Introduction: should be edited removing the paragraph and linking the different topic presented in order to lead the reader through the literature and towards the aim of the paper. Also some section with detailed description of materials properties (e.g. nanocrystals degradation) should be avoided.

Response 1: The introduction has been reorganized and excessive details have been removed for the sake of brevity. The progression entails electrospinning>ES applications>PVA and some applications>Post treatment of fibers>CNC and information>Thermal analysis of CNC/PVA electrospun composites>PVA/CNC composites, solutions and electrospinning>objective

Point 2: The goal of the manuscript should be clearly stated in the introduction

Response 2: Goal is stated in the last paragraph of the introduction

Point 3: The novelty of the work should be clearly discussed with respect to the mentioned references.

Response 3: An explanation of the novel aspect was included in the last paragraph of the introduction of the paper.

Point 4: Figure 1: it would be nice to see micrographs at higher magnification.

Response 4: There was not enough time to take more micrographs at higher resolutions for the mats.

Point 5: TGA: the TG and dTG of neat CNC should be reported. The authors should evaluated the residues of the two components at high temperatures in order to see if synergistic or simply additive effects occur during degradation (the use of calculated vs experimental tg plot is suggested). The weight loss of sample 7.20 in the 350-500 range is strange with respect to the other samples can the authors explain this ?

Response 5: Statements were added pointing the reader towards the Peng et al. 2012 study in which the CNC was thermally characterized. This material was obtained from the same source as the materials used in this study. Thermal information should be theoretically similar if not the same. The 80PVA20CNC grouping was abnormal after reviewing data. A more regular response was included in the reported material. It’s appears that the 20% CNC fill does have some synergistic effect because the PVA and CNC does not respond similarly, as in the 50% CNC samples.

Point 6: The discussion section is more like a short review paragraph of other works and should be adjusted so that the main results of the paper a discussed and comparison with other paper is performed.

Response 6: The discussion section was deleted and further discussion was outlined in the 3. Results and Discussion section.

Point 7: The conclusion paragraph should be ameliorated as it only contains a summary of the results achieved. 

Response 7: Values and some explanation were added to the conclusion of this paper.

Reviewer 2 Report

The aim of this study was to prepare poly (vinyl alcohol, PVA) and cellulose nanocrystals (CNC) random composite mats using the electrospinning method targeting plausible conditions for outside of the laboratory setting. The produced fibers were characterised using SEM and TGA methods. Mechanical properties along with viscosity and rheology measurements were also examined. The CNC fibers also increased mat stiffness and reduced strain to yield in non-treated mats. The use of CNCs demonstrates likely for compounding into bulk polymer composites as reinforcement filler and also shows potential for chemical crosslinking attributable to the –OH groups on both the PVA, in addition to esterification of the vinyl group, and CNC.

The paper is interesting; however the authors need to address the issues listed below before re-submission of this paper.

Major issues:

1.      The authors should discuss how influence of the distance between needle and target has on the morphology and nanofiber dimension.

2.      The charge of the deposited fibers has influence of the morphology of the fibers but there are other parameters such as solvent, which also contribute. The authors should make some comments related to the solvent factor on morphology of electrospun fibers.

Author Response

Response to Reviewer 2 Comments

Point 1: The authors should discuss how influence of the distance between needle and target has on the morphology and nanofiber dimension.

Response 1: This aspect of the study has been characterized and was not the focus of the article. A statement was included in the methods section pertaining to the use of a 15 cm TCD.

Point 2: The charge of the deposited fibers has influence of the morphology of the fibers but there are other parameters such as solvent, which also contribute. The authors should make some comments related to the solvent factor on morphology of electrospun fibers.

Response 2: This statement was included in the 2.2 solution preparation section. DI water is a simple to use, biologically safe solvent and did not negatively affect fiber morphology.

Reviewer 3 Report

This work introduces the fabrication of electrospun nanofibers mats of PVA embedded with CNCs. I do not recommend to publish the paper in the current form due to the following reasons:

1- The manuscript is not well-organized. For example, the objective should be mentioned by the end of introduction part and not in the methods section. Both abstract and conclusions miss numerical values about the outcome results.

2- The topic is not new and the authors did not clarify in a clear way what is the new thing that they introduced in this paper compared to other literature.

3- Samples codes have to be better clarified.

4- CNC nanostructures have to be imaged alone to prove that it is in nanoscale range.

5- Equations should have no caption. It should be removed.

6- Mechanical analysis curves "stress-strain" have to be shown in the manuscript.

7- Why did the authors use this range of KV? Why did not try higher ranges up to 25kV, which might lead to smaller diameter of PVA nanofibers instead of the shown semi-micro range?

8- The vision of showing viscousity measurements are not clear. Why and what is the importance?

9- References are not written on mdpi formatting. Also, the references have to be mentioned in numbers within the body of the paper "not the authors names". This is common in mdpi papers.

10- Lot of typos "may be due to different word/pdf processors".

Author Response

J. Elliott Sanders

Nanomaterials Reviewer Responses

Dear Editor

Response to Reviewer 3 Comments

Point 1: The manuscript is not well-organized. For example, the objective should be mentioned by the end of introduction part and not in the methods section. Both abstract and conclusions miss numerical values about the outcome results.

Response 1: The objective was moved to the end of the introduction section and better clarified by restating the target applications as well as the use of only summary statistics for analysis.

Point 2: The topic is not new and the authors did not clarify in a clear way what is the new thing that they introduced in this paper compared to other literature.

Response 2: Novelty was mentioned in the introduction of the literature with reference to the methods used and the amount of CNC filler used in this study.

Point 3: Samples codes have to be better clarified.

Response 3: A sample code section (2.1) was added to the methods section to clarify the intent of the sample code structure. It was slightly simplified; heat treated samples are expressed with an H and the non-treated samples have no symbol.

Point 4: CNC nanostructures have to be imaged alone to prove that it is in nanoscale range.

Response 4: CNC could not be individually imaged for the purposes of this study. However, the material was imaged in the Peng et al 2012 study and can be used for reference since the material was obtained from the same manufacturer (Forest Products Laboratory, Madison, WI).

Point 5: Equations should have no caption. It should be removed.

Response 5: Removed the caption explaining the equation

Point 6: Mechanical analysis curves "stress-strain" have to be shown in the manuscript.

Response 6: Stress-strain curves were added to the manuscript.

Point 7: Why did the authors use this range of KV? Why did not try higher ranges up to 25kV, which might lead to smaller diameter of PVA nanofibers instead of the shown semi-micro range?

Response 7: Literature suggests that a lower kV may lead to smaller fiber sizes. However, given the feed rate used in this study and the higher viscosity a 19 kV range was necessary to reduce droplet defects in spun mats. Lower kV could be useful to produce smaller fibers especially with the addition of CNC as it also reduced fiber diameter in the referenced literature.

Point 8: The vision of showing viscousity measurements are not clear. Why and what is the importance?

Response 8: Added a paragraph under viscosity that explains the significance of reporting viscosity data in this study.

Point 9: References are not written on mdpi formatting. Also, the references have to be mentioned in numbers within the body of the paper "not the authors names". This is common in mdpi papers.

Response 9: References were changed to nanomaterials formatting in Mendeley.

Point 10: Lot of typos "may be due to different word/pdf processors".

Response 10: Worked on misspellings and awkward sentences. Some wording is not in the Word processing dictionary like Thermogravimetric.

Reviewer 4 Report

This paper deals with the materials properties of the prepared PVA/CNC nanofiber by electrospinning. In this study, the composite fibers were prepared by electrospinning using PVA/CNC solution and composite materials were prepared by heat treatment.

The name and affiliation of the authors should be in accordance with the format of the journal.

Author should write the references according to the journal format and fill in the numbers in order.

Qualitative evaluations such as FT-IR should be added to the chemical structure or chemical interactions.

As shown in figure 1, in the case of the heat treated sample, the diameter of the fiber seems to be thicker, and an analysis related to the average diameter and distribution is considered necessary.

Figure 1 does not have a scale bar, so authors need to add it.

In case of mechanical properties, the mechanical properties are enhanced when the CNC is 20 % w/w. Why?

The properties after heat treatment are superior to those of heat treatment, but I wonder if there is a particular reason for heat treatment.

Author Response

J. Elliott Sanders

Nanomaterials Reviewer Responses

Dear Editor

Response to Reviewer 4 Comments

Point 1: The name and affiliation of the authors should be in accordance with the format of the journal.

Response 1: The author section was revised and another was added.

Point 2: Author should write the references according to the journal format and fill in the numbers in order.

Response 2: The references were organized according to the nanomaterials Mendeley plugin.

Point 3: Qualitative evaluations such as FT-IR should be added to the chemical structure or chemical interactions.

Response 3: FT-IR data will be included in subsequent studies.

Point 4: As shown in figure 1, in the case of the heat treated sample, the diameter of the fiber seems to be thicker, and an analysis related to the average diameter and distribution is considered necessary.

Response 4: Another image was selected to adequately show the fibers relative to the scale of the other micrographs; more fiber diameter measurements were taken on each mat produced for this study. ImageJ software was used to analyze each micrograph, setting the scale to the bar provided in the image, and the summary statistics were calculated using the software. Two mats per sample group (ex: 100PVA or 50PVA50CNCH) were used to measure diameter. Markers were left on the SEM micrograph to ensure the nano-fibers weren’t measured twice. For each mat 40 measurements were taken and averaged. Another average between the two mats was taken for each regular and heat treated sample groups.

Point 5: Figure 1 does not have a scale bar, so authors need to add it.

Response 5: Scale bars were cut short due to table formatting. The tables were adjusted and the scale bars are completely visible.

Point 6: In case of mechanical properties, the mechanical properties are enhanced when the CNC is 20 % w/w. Why?

Response 6: Stiffness was increased in 100PVA and 50PVA50CNC samples. However, the 80PVA20CNC samples showed better thermal resistance and elongation compared to untreated sample groups. Stress strain graphs show the mechanical strength response from increased CNC. Notes were made in the results and conclusion sections regarding these points.

Point 7: The properties after heat treatment are superior to those of heat treatment, but I wonder if there is a particular reason for heat treatment.

Response 7: A reason is discussed in the TGA and Mechanical properties sections of the results.

Round  2

Reviewer 1 Report

I’ve seen the authors tried to improve the manuscript by answering my queries. The conclusion should also report possible implications of the achieved results in terms of properties/applications.

Author Response

Response to Reviewer 1 Comments

Point 1: I’ve seen the authors tried to improve the manuscript by answering my queries. The conclusion should also report possible implications of the achieved results in terms of properties/applications.

Response 1: The conclusion was edited to explain where the results would be useful in context of the applications.

Reviewer 2 Report

Although the influence of the distance between needle and target has on the morphology and nanofiber dimension were not part of their primary research  goals, the authors need to address this in the revised manuscript. For example, they could comment their experimental set up with the similar set ups in literature.

Author Response

Response to Reviewer 2 Comments

Point 1: Although the influence of the distance between needle and target has on the morphology and nanofiber dimension were not part of their primary research goals, the authors need to address this in the revised manuscript. For example, they could comment their experimental set up with the similar set ups in literature.

Response 1: References to sources with the same or similar TCD were included with a statement in the methods section of the article.

Reviewer 3 Report

Most of the comments were addressed, but the stress-strain curves are not well described. It should be between [stress "F/A" not load] and [strain "elongation/original length not elongation only". Also, I still insist  on the importance of an image for CNC nanostructures before embedded as a filler or you should refer clearly in the reference that an image is refered to a reference (x) for the same manufacturer with exactly same specs of the sample.

Author Response

Response to Reviewer 3 Comments

Point 1: Most of the comments were addressed, but the stress-strain curves are not well described. It should be between [stress "F/A" not load] and [strain "elongation/original length not elongation only". Also, I still insist  on the importance of an image for CNC nanostructures before embedded as a filler or you should refer clearly in the reference that an image is referred to a reference (x) for the same manufacturer with exactly same specs of the sample.

Response 1: Stress-strain curves have been modified and mentioned in the results section of the article. A reference to the paper and the figure therein of the SEM micrograph of pictured CNC was included in the methods section.